# Inhibitory Effects of *Grewia tomentosa* Juss. on IgE-Mediated Allergic Reaction and DNCB-Induced Atopic Dermatitis

**DOI:** 10.3390/plants11192540

**Published:** 2022-09-27

**Authors:** Hwa Pyoung Lee, Wooram Choi, Ki Woong Kwon, Long You, Laily Rahmawati, Van Dung Luong, Wonhee Kim, Byoung-Hee Lee, Sarah Lee, Ji Hye Kim, Jae Youl Cho

**Affiliations:** 1Department of Integrative Biotechnology, Sungkyunkwan University, Suwon 16419, Korea; 2Department of Biology, Dalat University, 01 Phu Dong Thien Vuong, Dalat 670000, Vietnam; 3National Institute of Biological Resources, Environmental Research Complex, Incheon 222689, Korea

**Keywords:** *Grewia tomentosa* Juss., allergic inflammation, mast cell, passive cutaneous anaphylaxis, atopic dermatitis

## Abstract

*Grewia tomentosa* Juss. is a deciduous shrub that mainly grows in Asia. Despite studies of other *Grewia* species for treatment of various diseases, *Grewia tomentosa* Juss. has not been studied as a medicinal herb. This study evaluates the anti-allergic and anti-topic dermatitis activity of *Grewia tomentosa* Juss. ethanol extract (Gt-EE). The results show that Gt-EE suppressed IgE–antigen-induced β-hexosaminidase release. The mRNA expression of IL-1β, IL-4, IL-5, IL-6, IL-13, TNF-α, MCP-1, and TSLP, which are involved in allergic responses, was inhibited by Gt-EE in IgE-stimulated RBL-2H3 cells. In addition, the phosphorylation of Syk, PLCγ1, PKCδ, PI3K, AKT, NF-κB p65, NF-κB p50, p38, JNK, and ERK1/2 was decreased by Gt-EE in these cells. Gt-EE also showed anti-inflammatory effects in in vivo mouse models. In passive cutaneous anaphylaxis (PCA), a commonly used mouse model, Gt-EE decreased the allergic response, infiltration of mast cells, and mRNA level of IL-4. Furthermore, Gt-EE ameliorated symptoms of DNCB-induced atopic dermatitis (AD). In DNCB-induced AD, Gt-EE suppressed the increase in mast cells, serum IgE level, expression of allergic mediators (IL-1β, IL-4, IL-5, IL-6, TNF-α), and phosphorylation of proteins (IκBα, NF-κB p65, NF-κB p50, p38, JNK, and ERK1/2) implicated in allergic reactions

## 1. Introduction

The prevalence of allergic diseases is increasing worldwide, and those diseases reduce the quality of life of affected patients. Allergic disorders include allergic rhinitis, asthma, drug hypersensitivity, atopic dermatitis, urticaria, food allergy, eczema, life-threatening anaphylaxis, conjunctivitis, eosinophilic esophagitis, angioedema, hypersensitivity pneumonitis [1]. Globally, 400 million people have rhinitis, and 300 million people suffer from asthma [2]. Allergic reactions occur when allergen-responsive type 2 helper T cells (Th2) induce the activation or recruitment of immunoglobulin E (IgE) by antibody-producing B cells, eosinophils, and mast cells [3]. Allergens that enter the body through the respiratory tract, alimentary canal, or skin are perceived by B cells or macrophages. The allergens are fragmented and displayed on the cell surfaces for interaction with Th2 cells. Th2 cytokines help recruit B cells, eosinophils, and mast cells, all of which can produce IgE antibodies. This process plays a pivotal role in the induction of allergic symptomatology [4]. In particular, mast cells serve as key effectors of IgE-mediated allergic and inflammatory reactions, including allergic rhinitis, atopic dermatitis, and asthma [5]. The high-affinity IgE receptor FcεRI is expressed on the surfaces of mast cells, and crosslinked IgE–FcεRI complexes are induced when a multivalent antigen binds to IgE bound to FcεRI [6]. Activation of the IgE/FcεRI pathway gives rise to mast cell degranulation and promotes the secretion of mediators that induce allergic reactions [7]. Because the secretion of these mediators also leads to inflammation, a method to inhibit the activation of mast cells could help to treat allergic symptoms.

Atopic dermatitis (AD) is a chronically relapsing inflammatory skin disease characterized by severe itching, erythema, unbalanced immune system, and eczematous skin lesions [8]. Worldwide, the prevalence rate of AD in children is 15% to 20%; in adults, it is 1% to 3% [9]. Because of its symptoms, AD can increase behavioral disorders, anxiety, depression, and attention-deficit/hyperactivity disorder, significantly reducing the patient’s overall quality of life and even adversely affecting their family. Therefore, appropriate treatments must be found [10]. The typical feature of AD is the biased differentiation of naïve T cells into Th2 cells. Among the various mediators generated by a Th2-biased immune response, interleukin (IL)-4, which induces IgE production, is overexpressed in AD patients. As mentioned above, upon encountering an antigen, IgE-bound FcεRI on mast cells becomes aggregated, which leads to cell activation [11]. FcεRI aggregation results in phosphorylation of immunoreceptor tyrosine-based activation motifs (ITAMs), which are located in the cytoplasm of FcεRI. Phosphorylated ITAMs provide a docking site for the signal-propagating kinase Syk, which is activated and autophosphorylated by binding to ITAMs [12]. Syk phosphorylation leads to activation of numerous signaling molecules, including phospholipase Cγ (PLCγ) and phosphatidylinositol 3-kinase (PI3K) [13]. Activated PLCγ triggers the phosphorylation of protein kinase Cδ (PKCδ), which has been implicated in degranulation of RBL-2H3 mast cells [14,15]. In mast cells, phosphorylated PI3K induces cytokine production by activating AKT and nuclear factor kappa B (NF-κB), including p65 and p50 [16]. In addition, FcεRI–IgE crosslinking activates mitogen-activated protein kinases (MAPKs), including p38, extracellular signal-regulated kinase (ERK), and c-Jun N-terminal protein kinase (JNK), which leads to generation of lipid mediators such as leukotrienes and prostaglandins [17].

*Grewia tomentosa* Juss., also known as *Grewia hirsuta* Sm., is a deciduous shrub belonging to the family Malvaceae [18]. *Grewia tomentosa* Juss. can grow quickly, survive in wet soil, and is resistant to shade, so it can grow on a forest floor covered by a canopy [19]. It is distributed in Malaysia, Thailand, Cambodia, Jawa, Myanmar, Sumatera, and Lesser Sunda Island. *Grewia* species have traditionally been used as medicinal herbs for various diseases. *Grewia hirsuta* Vahl has been reported to cure ulcers [20], rheumatism, headaches, sores, cholera, joint pain [21], and diarrhea [21,22]. *Grewia asiatica* Linn. has been used to cure sore throats, colds, coughs, asthma, and bronchitis, as well as skin problems such as eruptions, eczema, and inflammation [23]. *Grewia tenax* has been reported to treat intestinal infections, dysentery, hepatic disorders, jaundice, fever, diarrhea, rheumatism, and distress of the stomach and skin [24]. The pharmaceutical uses of *Grewia* species are also supported by recent scientific studies. *Grewia hirsuta* Vahl has been reported to possess analgesic, anti-inflammatory, anti-diarrheal, antimicrobial [25], antioxidant and antiproliferative properties [26]. The antioxidant, antimicrobial [27], antimalarial, antiemetic and antidiabetic [28] activity of *Grewia asiatica* Linn. has also been revealed. Despite various pharmacological studies on *Grewia* species, no studies have been conducted on *Grewia tomentosa* Juss.

Therefore, in this work, we evaluated the effect of an ethanol extract of *Grewia tomentosa* Juss. (Gt-EE) on IgE/antigen-stimulated mast cells (RBL-2H3) and a passive cutaneous anaphylaxis (PCA) mouse model. In addition, we investigated the effect of Gt-EE on a 2,4-dinitrochlorobenzene (DNCB)-induced mouse model of AD.

## 2. Results

### 2.1. Anti-Allergic Activities of Gt-EE

Because of the RBL-2H3 cell line, an analogue of mast cells is suitable for studying the mechanisms of mast cell–mediated allergic reactions, and we used it to investigate the anti-allergy efficacy of Gt-EE [29]. First, the cytotoxicity of Gt-EE to RBL-2H3 cells was evaluated by a cell viability assay. When RBL-2H3 cells were treated with 100 μg/mL Gt-EE for 24 h, the viability was 94.4%, indicating non-cytotoxicity (Figure 1A). RBL-2H3 cells are frequently used in in vitro studies of degranulation [30]. After sensitizing RBL-2H3 cells with anti-dinitrophenol (DNP) IgE, treating those cells with DNP conjugated with human serum albumin (HSA) induces degranulation [31]. The degranulation of RBL-2H3 cells can also be induced by synthetic compounds such as calcium ionophore (A23187) and compound 48/80, which are used as a simple method for studying the mechanisms of allergic responses in vitro [32,33]. β-Hexosaminidase is an enzyme released when RBL-2H3 cells are activated and can be used to quantify the degree of degranulation of variously stimulated RBL-2H3 cells [34]. We confirmed the anti-allergy effects of Gt-EE (i.e., confirmed that Gt-EE inhibited the degranulation of RBL-2H3 cells) in terms of β-hexosaminidase release activity. First, when degranulation of RBL-2H3 cells was induced by anti-DNP IgE and DNP-HSA, Gt-EE inhibited the allergic reaction in a dose-dependent manner (Figure 1B). Similarly, when RBL-2H3 cells were treated with A23187 and phorbol 12-myristate 13-acetate (PMA) or compound 48/80, Gt-EE dose-dependently inhibited the degranulation of RBL-2H3 cells (Figure 1C,D).

To determine which component contributes to the anti-allergy effect of Gt-EE, the phytochemical components were analyzed using gas chromatography–mass spectrometry (GC-MS). A primary compound was *n*-hexadecanoic acid, also known as palmitic acid (Figure 1E), which has a 16-carbon backbone and is a commonplace saturated long-chain fatty acid. Palmitic acid is found in human bodies and can be provided by diet (it is a major component of palm oil) and endogenous synthesis from other carbohydrates, fatty acids, and amino acids [35]. Palmitic acid has been reported to have several pharmacological activities. For example, it plays an important role in the biological defense against pathogenic microorganisms by contributing to IgA production in the intestine [36]. It has also been reported to have anti-allergy activity because it significantly inhibited A23187- or antigen-induced degranulation in RBL-2H3 cells [37]. The other compounds in Gt-EE, including lupeol and friedelan-3-one (friedelin), that have been reported to have anti-inflammatory activity are listed in Table 1 [38,39].

Additionally, we conducted an enzyme activity assay. Lipoxygenases (LOXs) are dioxygenases that catalyze the oxygenation of polyunsaturated fatty acids such as linoleic acid and arachidonic acid [40]. LOX enzymes produce leukotrienes from arachidonic acid, and those leukotrienes are implicated in several inflammatory and allergic disorders, including skin inflammatory disorders, chronic obstructive pulmonary disorder, and bronchial asthma [41]. Therefore, LOX inhibition has been regarded as a potential way to treat those diseases. To evaluate the bioactivity of Gt-EE against LOXs, the 15-lipoxygenase inhibitory assay was carried out. As shown in Table 2, Gt-EE inhibited the activity of 15-lipoxygenase by 34.6% and 43.6% at concentrations of 50 and 100 μg/mL, respectively. Because quercetin is a well-known competitive inhibitor of LOX, it was used as a positive control [42].

### 2.2. Effects of Gt-EE on the mRNA Expression of Allergic Response–Related Cytokines and the Activation of the IgE–FcεRI Signaling Pathway

Antigen exposure to IgE-sensitized RBL-2H3 cells leads to production and secretion of diverse cytokines and chemokines that cause allergic inflammation. To determine whether Gt-EE downregulates that allergic response, we evaluated the mRNA expression levels of IL-1β, IL-4, IL-5, IL-6, IL-13, TNF-α, MCP-1, TSLP, and TGF-β1 in IgE-stimulated RBL-2H3 cells. Anti-DNP IgE–sensitized RBL-2H3 cells were treated with Gt-EE for 30 min, stimulated with DNP-HSA for 24 h, and harvested to analyze the mRNA levels of those cytokines. As shown in Figure 2A,B, the mRNA expression levels of the cytokines were increased by DNP-HSA stimulation, and all but TGF-β1 were decreased in the group treated with Gt-EE. Like the other cytokines, the mRNA expression of TGF-β1 was increased by stimulation with IgE and antigen, and its mRNA level was further increased by Gt-EE (Figure 2A,B).

Crosslinked IgE–FcεRI complexes activate intracellular signaling pathways that lead to cytokine secretion and degranulation of mast cells. We performed an immunoblotting assay to determine the efficacy of Gt-EE on several of the signaling pathways related to allergic reactions. As indicated in Figure 2C–E, stimulating RBL-2H3 cells with IgE induced the phosphorylation of Syk, PLCγ1, PKCδ, PI3K, AKT, NF-κB p65, NF-κB p50, p38, JNK, and ERK1/2. However, Gt-EE treatment markedly inhibited the phosphorylation of the molecules implicated in those allergic response–related signaling pathways (Figure 2C–E).

### 2.3. Gt-EE Attenuated IgE–Antigen-Induced Passive Cutaneous Anaphylaxis

To research drugs with potential anti-allergic effects, the in vivo passive cutaneous anaphylaxis (PCA) model has been organized using anti-DNP IgE antibodies and antigens [29]. We induced topical and immediate hypersensitivity reactions in the ears of BALB/c mice and then monitored the quantity of Evans blue dye taken out from those ear tissues. The scheme of the experiment is summarized in Figure 3A. The ear tissues showing the passive skin hypersensitivity reaction were stained quickly and intensely by Evans blue dye, but the ears of the group pretreated with orally administered Gt-EE were less heavily stained (Figure 3B,C). In addition, oral administration of Gt-EE improved the ear swelling caused by allergic inflammation (Figure 3B,D). Allergic inflammation increases the number of the mast cells at affected sites [43]. The toluidine blue O staining results indicate that oral administration of Gt-EE inhibited the infiltration of mast cells (Figure 3B,E). During allergic inflammation, cytokine production by the activated mast cells also increases, along with the increase in mast cell infiltration. As shown in Figure 3F, IL-4, the main cytokine associated with IgE-mediated allergic reactions, decreased in the Gt-EE group.

### 2.4. Gt-EE Ameliorated DNCB-Induced Atopic Dermatitis

DNCB is a representative irritant used for research because it induces AD-like skin damage by repeated application [44]. The experimental schedule is summarized in Figure 4A. Repeated exposure to DNCB made the mouse ear tissues thick, rough, and red. However, concurrent application of Gt-EE lessened these AD symptoms (Figure 4B). The dermatitis score was determined using a sensory evaluation that considered itching, erythema, and dryness. In the DNCB solo group, the dermatitis score was significantly higher than in the normal group, and it was significantly decreased by application of Gt-EE (Figure 4C). The ear tissue that was considerably thickened by AD was thinner in the Gt-EE groups (Figure 4B,D). The water content of the ear tissue was measured to assess the dryness of the AD-like lesions, but neither that nor body weight differed significantly between the groups (Figure 4E,F). The number of mast cells in the ear tissue was increased by AD, and Gt-EE significantly ameliorated that increase (Figure 4B,G). Because elevated IgE production is one of the key features of Th2 immune responses, the serum IgE level, a diagnostic marker of AD, was determined [45]. The amount of IgE in the serum of the AD group was higher than normal, but Gt-EE restricted it to the normal level (Figure 4H).

### 2.5. Inhibitory Effects of Gt-EE on mRNA Expression of Cytokines and Activation of Signaling Pathways in AD-like Lesions

In the worsening of AD caused by DNCB treatment, the expression of allergic inflammation-related cytokines increases, and the NF-κB and MAPK signaling pathways are activated [46]. Using ear tissues in which AD had been induced, we evaluated whether Gt-EE inhibits the expression of those cytokines and activation of those signaling pathways. Although the mRNA expression of IL-1β, IL-4, IL-5, IL-6, and TNF-α in ear tissues with AD was significantly higher than normal, those levels were significantly restricted by Gt-EE treatment (Figure 5A–E). In addition, Gt-EE application suppressed the phosphorylation of IκBα, NF-κB p65, NF-κB p50, JNK, and ERK1/2 in DNCB-induced AD-like lesions (Figure 5F,G).

## 3. Discussion

Type 1 hypersensitivity is caused by secretion of allergy-related mediators, proinflammatory cytokines, and chemokines from activated mast cells [47]. After antigen exposure, mast cells are stimulated by crosslinked IgE–FcεRI complexes. The crosslinking leads to release of allergic mediators that trigger allergic responses such as asthma, AD, conjunctivitis and allergic rhinitis [48]. Therefore, mast cells are commonly considered targets to treat allergic symptoms. RBL-2H3 cells of rat basophilic leukemia are analogues of mast cells which have highly expressed FcεRI, and like mast cells, they can be stimulated by IgE–antigen complexes [49].

In this study, we examined the anti-allergy effects of an ethanol extract of *Grewia tomentosa* Juss. on IgE-stimulated RBL-2H3 cells, a PCA mouse model, and a DNCB-induced AD mouse model. Gt-EE suppressed the degranulation of RBL-2H3 cells caused by IgE/antigen, PMA/A23187, or compound 48/80 (Figure 1). In addition, Gt-EE inhibited the mRNA expression of cytokines implicated in allergic responses (Figure 2A,B). Activation of mast cells leads to the synthesis and secretion of various cytokines that continue the inflammatory response. Cytokines such as IL-1β, IL-4 IL-5, IL-6, IL-13, TNF-α, MCP-1, and TSLP induce allergy-related inflammation, which causes tissue fibrosis, granuloma formation, and leukocyte infiltration. IL-1β is a typical proinflammatory cytokine that is implicated in many inflammatory conditions, including allergic and autoinflammatory disorders such as AD, bronchial asthma, and contact hypersensitivity [50]. IL-4 is crucial in the development of allergic disorders because of its influence over T helper cell development and production of IgE [51]. IL-4 also promotes the expression of IL-5, which is a cytokine responsible for maturation and release of eosinophils [52]. During the acute stage of inflammation, IL-6 plays a central role in exacerbation of Th2-mediated diseases, including asthma and allergic airway inflammation [53]. The Th2 cytokine IL-13, with IL-4, is a crucial regulator of IgE generation [54]. TNF-α plays a pivotal role in the pathogenesis of allergies and has been reported to contribute to both the early and late phases of allergy development [55]. MCP-1 is a chemokine responsible for regulating migration and infiltration of various immune cells into inflamed sites at the beginning of the immune response, and it also activates the basophils that are attracted to sites of inflammation. It promotes the production of IL-4 to induce differentiation of Th2 cells and directly induces degranulation of pulmonary mast cells [56]. TSLP is crucial for differentiating dendritic cell–mediated CD4+ T-cells into Th2 cells, and it is closely related to allergic diseases such as asthma, AD, allergic rhinoconjunctivitis, and eosinophilic esophagitis [57]. Although some studies have indicated that TSLP is expressed in mast cells [5,58], it is mainly produced in epithelial cells [59]. TSLP secreted from epithelial cells can activate mast cells to release proinflammatory cytokines [60]. Therefore, TSLP is closely associated with AD. An immunohistochemical analysis found TSLP overexpression in keratinocytes from both acute and chronic lesions in AD patients [61]. Moreover, TSLP overexpression in keratinocytes exacerbated AD by increasing the infiltration of Th2-related cells and the serum IgE level [62]. In this study, we demonstrated that Gt-EE decreased the mRNA expression level of TSLP in mast cells (Figure 2B). However, because TSLP is mainly produced and secreted by epithelial cells, further studies are needed to determine whether Gt-EE inhibits the production and secretion of TSLP in epithelial cells. Gt-EE upregulated the expression of TGF-β1 mRNA in allergic conditions (Figure 2B). TGF-β1 can suppress IgE-mediated mast cell activation by inhibiting the synthesis of IgE and the proliferation of mast cells [63]. TGF-β1 is also known to inhibit the expression of FcεRI in the IgE-mediated allergic response of mast cells [64]. Furthermore, in the presence of TGF-β1, proinflammatory mediators such as histamine and leukotriene secreted by mast cells are reduced [65]. TGF-β1 inhibits IgE secretion from activated B cells [66], and the role of TGF-β1 produced by Treg cells has been known to inhibit allergic reactions [67]. We demonstrated that, during an allergic response, Gt-EE promoted the production of TGF-β1 mRNA, which had already been increased in mast cells. Therefore, this suggests that Gt-EE could induce the production of cytokines (eg., TGF-β), leading to suppression of allergic reactions.

Leukotrienes (LTs) are well known mediators involved in the pathogenesis of allergic diseases such as asthma, allergic rhinitis, AD, and chronic urticaria [68]. Upon stimulation with a specific antigen, mast cells produce several eicosanoids, including LTs. The production of eicosanoids is triggerd by phosphorylation and activation of cytoplasmic phospholipase A_2_ (cPLA_2_) via an ERK-dependent pathway [69]. The activated cPLA_2_s liberate arachidonic acids from the plasma membrane, and then those free arachidonic acids are metabolized to LTA_4_ by 5-LOXs. The subsequent conversion of LTA_4_ to LTB_4_, LTC_4_, and LTD_4_ can then occur [70]. In our lipoxygenase enzyme inhibitory assay, Gt-EE inhibited lipoxygenase (Table 2). Interestingly, Gt-EE also downregulated the phosphorylation of PKCδ, which has been reported to be involved in IgE signaling to ERK and cPLA_2_ phosphorylation (Figure 2C) [71]. In addition, the level of ERK phosphorylation was reduced by Gt-EE both in vitro and in vivo (Figure 2E and Figure 5G). These results suggest that Gt-EE could have synergistic effects that inhibit the synthesis of LTs.

Gt-EE ameliorated the inflammatory reactions of type I allergy in our PCA mouse model (Figure 3). It also improved DNCB-induced AD (Figure 4 and Figure 5). Because increase in mast cell infiltration leads to an increase in the secretion of Th2 cytokines by mast cells [72], inhibition of mast cell infiltration by Gt-EE appears to mitigate the symptoms of PCA and AD. In particular, considering the importance of IL-4 in allergic reactions, it is noteworthy that Gt-EE reduced the mRNA expression of IL-4 in both the PCA and AD models (Figure 3F and Figure 5B). Production of allergic inflammatory cytokines, including IL-4, occurs through the NF-κB and MAPK signaling pathways [73]. Gt-EE inhibited the phosphorylation of NF-κB p65, NF-κB p50, p38, JNK, and ERK1/2 in vitro and also suppressed the phosphorylation of NF-κB p65, NF-κB p50, JNK, and ERK1/2 in vivo (Figure 2D,E and Figure 5F,G). Collectively, these results suggest that Gt-EE ameliorates allergic reactions and has potential for use as a new therapeutic herb.

## 4. Materials and Methods

### 4.1. Materials

RBL cells (rat basophilic leukemia) were obtained from the American Type Culture Collection (ATCC) (Rockville, MD, USA). Sodium dodecyl sulfate (SDS), dimethyl sulfoxide (DMSO), DNP IgE, DNP-HSA, calcium ionophore A23187, PMA, compound 48/80, 4-nitrophenyl-N-acetyl-β-D-glucosaminide, lipoxygenase from *Glycine max* (soybean), linoleic acid, (3-4,5-dimethylthiazol-2-yl)-2,5-diphenyl-tetrazolium bromide (MTT), Evans blue, toluidine blue O, DNCB, and formaldehyde solution were bought from Sigma (St. Louis, MO, USA). Trypsin (0.25%) was purchased from HyClone Laboratories (Logan, UT, USA). Additionally, TRI Reagent^®^ solution was acquired from Molecular Research Center, Inc. (Cincinnati, OH, USA). 1X phosphate-buffered saline (PBS) was purchased from Samchun Pure Chemical Co. (Gyeonggi-do, Korea). The sets of primers for quantitative real-time polymerase chain reaction (PCR) were synthetized by Macrogen (Seoul, Korea). Horse anti-mouse HRP-conjugated secondary antibody, goat anti-rabbit HRP-conjugated secondary antibody, and the antibodies against the total and phosphorylated forms of Syk, PLCγ1, PKCδ, PI3K, AKT, NF-κB p65, p38, JNK, and ERK1/2 were acquired from Cell Signaling Technology (Beverly, MA, USA), and those against NF-κB p50 and β-actin were purchased from Santa Cruz Biotechnology, Inc. (Dallas, TX, USA).

### 4.2. Cell Culture

RBL-2H3 cells were cultured as monolayers in Dulbecco’s Modified Eagle’s Medium (DMEM) (HyClone Laboratories, Logan, UT, USA) supplemented with 10% heat-inactivated fetal bovine serum (FBS) (Gibco, Grand Island, NY, USA) and 1% penicillin/streptomycin (HyClone Laboratories, Logan, UT, USA). The cultured cells were maintained in a 5% CO2 humidified incubator at 37 °C.

### 4.3. Gt-EE Preparation

Gt-EE (Code number: NIBR 928) was obtained from the National Institute of Biological Resources (NIBR) (Incheon, Korea). The aerial parts of *Grewia tomentosa* Juss. were collected in Tịnh Biên, An Giang, Vietnam by D.C. Nguyen and V.M. Trinh on 23 August 2018 (Specimen No. NIBRVP0000722658). The preparation of Gt-EE was performed as previously reported [74]. Briefly, the dried branches with aerial parts of *Grewia tomentosa* Juss. were soaked in 70% ethanol and Gt-EE was prepared by extraction in an ultrasonic extractor (Ultrasonic Cleaner UC-10, UC-20, 400 W) for 3 h at 50 °C (three times). The extract was kept in a freezer compartment at −20 °C until use. For the in vitro studies, the dried Gt-EE stock was dissolved in DMSO to make a 100 mg/mL of Gt-EE stock solution. For each experiment, the Gt-EE stock solution was diluted to the desired final concentration of 25–100 μg/mL in a suitable culture medium. For the PCA and DNCB-induced AD mouse model experiments, the dried Gt-EE stock was dissolved in 0.5% carboxymethyl cellulose (CMC) and PBS, respectively, to doses of 50 and 100 mg/kg and 4 and 8 mg/kg, respectively. The doses of Gt-EE in this study were decided in accordance with previous papers covering similar in vitro and in vivo experiments [75,76].

### 4.4. Cell Viability Assay

RBL-2H3 cells were seeded in 96-well plates at a density of 5 × 10^4^ cells/mL. To investigate the cytotoxicity of Gt-EE, RBL-2H3 cells were treated with 25, 50, and 100 μg/mL of Gt-EE for 24 h. Cell viability was evaluated using the MTT assay. After discarding 100 μL of the cultured media, the cells were incubated with 10 µL/well of MTT solution (10 mg/mL in PBS, pH 7.4). After 3 h, the RBL-2H3 cells were treated with 100 µL of MTT stopping solution (10% sodium dodecyl sulfate with 10 mM HCl in distilled water) overnight, and then the absorbance of the solubilized formazan at 570 nm was detected using an optical density reader (BioTek Instruments Inc., Winooski, VT, USA).

### 4.5. β-Hexosaminidase Activity Assay

RBL-2H3 cells were seeded into 96-well plates (5 × 10^5^ cells/mL) in DMEM with 10% FBS and incubated overnight. RBL-2H3 cells were sensitized with anti-DNP IgE (100 ng/mL) at 37 °C overnight. After being washed 3 times with Siraganian buffer, the cells were treated with Gt-EE (25, 50, and 100 μg/mL) dissolved in Siraganian buffer for 30 min. Then, the cells were stimulated with DNP-HSA (1 μg/mL) for 24 h at 37 °C. The stimulation was stopped on ice for 5 min, and the cell supernatants (50 μL) were moved to new 96-well plates and incubated with 50 μL of 1 mM 4-nitrophenyl-N-acetyl-β-D-glucosaminide in 0.1 M sodium citrate (pH 4.5) at 37 °C for 1 h. The reaction was terminated with 200 μL/well of carbonate buffer containing 0.1 M Na_2_CO_3_ and 0.1 M NaHCO_3_ (pH 10), and the absorbance was measured at 405 nm using a microplate reader (BioTek Instruments Inc., Winooski, VT, USA). In addition, RBL-2H3 cells incubated in 96-well plates (5 × 10^5^ cells/mL) in DMEM with 10% FBS overnight were washed 3 times with Siraganian buffer and then treated with Gt-EE at concentrations of 25, 50, and 100 μg/mL for 30 min before stimulation with 1 μM A23187 and 50 nM PMA or 30 μg/mL compound 48/80 for 1 h at 37 °C. The stimulation was stopped on ice for 5 min, and then cell supernatants were collected for the β-hexosaminidase activity assay, as described above.

### 4.6. Gas Chromatography–Mass Spectrometry

A GC-MS analysis of Gt-EE was performed by the Cooperative Center for Research Facilities of Sungkyunkwan University (Gyeonggi-do, Korea), as previously described [77]. Briefly, GC was conducted using an Agilent 8890 GC instrument (Santa Clara, CA, USA) equipped with an Agilent J&W DB-624 Ultra Inert GC column (60 m in length × 250 μm in diameter × 1.40 μm in thickness), and MS was performed using an Agilent 5977B MSD instrument (Santa Clara, CA, USA) equipped with a Series II triple-axis detector with a high energy dynode and long-life electron multiplier. The spectrum of phytochemicals in the National Institute of Standards and Technology library was used to identify unknown phytochemicals in Gt-EE.

### 4.7. Inhibition of 15-Lipoxygenase Activity Assay

The 15-LOX inhibitory activity of Gt-EE was determined according to the method described by Yasin et al. with a slight modification [78]. In addition, 15-LOXs catalyze the reaction between linoleic acid and oxygen, producing 13-hydroperoxyoctadecadienoic acid that increases absorbance at 234 nm. To prepare the reaction mixture, 200 μL of sample (Gt-EE or quercetin) and 400 μL of soybean lipoxygenase solution (167 U/mL) were mixed in 3.2 mL of 100 mM sodium phosphate buffer (pH 7.4), and then the reaction mixture was incubated at 25 °C for 10 min. Quercetin was used as the positive control and prepared by omitting the sample from the mixture and adding only solvent (DMSO). The reaction was initiated by the addition of 200 μL of 2.5 mM sodium linoleic acid solution. Absorbance was measured at 234 nm every 30 s for 3 min using a UV-vis spectrophotometer (BioTek Instruments Inc., Winooski, VT, USA). The lipoxygenase inhibitory activity was then calculated using the following equation:Lipoxygenase inhibitory activity % = [(ΔA_1_/Δt − ΔA_2_/Δt)]/(ΔA_1_/Δt)] × 100
where A_1_/Δt is the enzymatic activity in the control, and A_2_/Δt is the enzymatic activity in the samples.

### 4.8. Quantitative Real-Time PCR

To measure the expression of genes related to the allergy response, RBL-2H3 cells were seeded in a 12-well plate at a density of 5 × 10^5^ cells/mL and treated with Gt-EE (50 and 100 μg/mL) for 24 h. Total RNA was extracted using TRI Reagent^®^ solution in accordance with the manufacturer’s instructions. Quantification of total RNA was carried out using a Take3 micro-volume plate (BioTek Instruments Inc., Winooski, VT, USA). We used 1 μg of total RNA to synthesize cDNA using a cDNA synthesis kit (Thermo Fisher Scientific, Waltham, MA, USA) according to the manufacturer’s instructions. mRNA expression levels were investigated by real-time PCR using 2x qPCRBIO SyGreen Blue Mix Lo-ROX in accordance with the manufacturer’s instructions (PCR Biosystems Ltd., London, UK) on a CFX96 real-time PCR detection system (Bio-Rad Laboratories Inc., Hercules, CA, USA). The amplification conditions for real-time PCR were as follows: 10 s denaturation time at 95 °C, 10 s annealing time at 58 °C, and 60 s extension time at 72 °C for 39 cycles, with detection of fluorescent product as the last step of each cycle. The primer sequences used in this experiment are listed in Table 3.

### 4.9. Preparation of Whole Cell Lysates and Immunoblotting Analysis

Gt-EE-treated RBL-2H3 cells were harvested with cold PBS using a cell scraper and then lysed for 30 min on ice in cell lysis buffer (20 mM NaF, 25 mM β-glycerol phosphate pH 7.5, 120 mM NaCl, 2% NP-40, 2 μg/mL aprotinin, 2 μg/mL leupeptin, 50 mM Tris-HCl pH 7.5, 100 μM Na3VO4, 2 μg/mL pepstatin A, 1 mM benzamide, 1.6 mM pervanadate, and 100 μM PMSF). The cell lysates were centrifuged at 12,000× *g* for 15 min at 4 °C to settle the cell debris and then stored at −70 °C until use. Protein concentrations were quantified using the Bradford assay (Bio-Rad, Hercules, CA, USA), and then immunoblotting analysis was carried out as previously described [79].

### 4.10. IgE-Mediated PCA Mouse Model

A total of 40 BALB/c mice (female, 8 weeks old) was randomly divided into 5 groups (PBS only, DNP-HSA only, DNP-HSA and Gt-EE 50 mg/kg, DNP-HSA and Gt-EE 100 mg/kg, and DNP-HSA and dexamethasone 10 mg/kg). Before induction of PCA, the BALB/c mice in the Gt-EE groups were treated with an oral administration of Gt-EE (50 or 100 mg/kg) dissolved in 0.5% CMC once a day for 5 days. To induce a PCA reaction, the ears of the mice were sensitized by an intradermal injection of anti-DNP IgE (0.5 µg/site). The next day, the mice were stimulated with an intravenous injection of DNP-HSA (1 mg/mouse) and 1% (*v/v*) Evans blue solution. After 30 min, the mice were sacrificed, and the dissected ears were soaked in 1 mL of formamide and left overnight at 65 °C for extravasation of the Evans blue dye. The absorbance of Evans blue dye was measured at 620 nm with a spectrophotometer (BioTek Instruments Inc., Winooski, VT, USA).

### 4.11. DNCB-Induced AD Mouse Model

A total of 25 BALB/c mice (male, 5 weeks old) was randomly divided into 5 groups (PBS only, DNCB only, DNCB and Gt-EE 4 mg/kg, DNCB and Gt-EE 8 mg/kg, and DNCB and dexamethasone 8 mg/kg) [78,79]. DNCB, Gt-EE, and dexamethasone were all dissolved in PBS and applied directly to the ears of BALB/c mice. Gt-EE and dexamethasone (200 μL/ear) were applied once a day for 28 days. On day 9, 1% DNCB (200 μL) was painted on each ear. Then, treatment with 0.5% DNCB was repeated every 3 days. On day 28, blood samples were collected after euthanasia. The serum was used in an ELISA kit (BD Biosciences, Oxford, UK) according to the manufacturer’s instructions to measure the serum IgE level. After blood collection, the ears were removed and used for histopathological analysis.

### 4.12. Histopathological Analysis

The ears of each mouse were fixed with 4% formaldehyde and embedded in paraffin. Four-μm-thick paraffin sections were prepared and stained with H&E for histological evaluation or with toluidine blue O for mast cell identification [80]. The number of mast cells in the ear tissues was manually counted from 10 random views under a light microscope at 400× magnification and is expressed as the count per high-power field.

### 4.13. Statistical Analysis

All data acquired from this study are presented as the mean ± standard deviation of at least three independent experiments. Statistical analyses were carried out using GraphPad Prism 8 statistics software (GraphPad Software, San Diego, CA, USA). All results were analyzed using Student’s *t*-test and one-way ANOVA followed by Dunnett’s test. A *p*-value < 0.05 was considered statistically significant.

## Figures and Tables

**Figure 1 plants-11-02540-f001:**
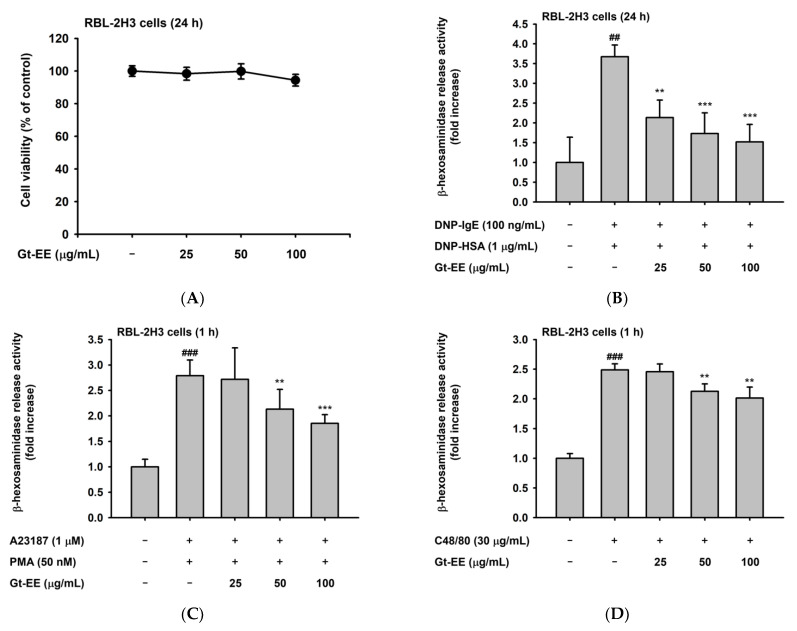
Inhibitory activity of Gt-EE on degranulation of RBL-2H3 cells and the phytochemical components of Gt-EE. (**A**) The cytotoxicity of Gt-EE to RBL-2H3 cells was determined using the MTT assay. RBL-2H3 cells were treated with Gt-EE for 24 h, and then cell viability was determined. (**B**–**D**) The degranulation of stimulated RBL-2H3 cells was investigated by establishing the amount of β-hexosaminidase released. (**B**) IgE-sensitized RBL-2H3 cells were treated with Gt-EE for 30 min and then challenged with DNP-HSA for 24 h. (**C**) RBL-2H3 cells were stimulated with PMA/A23187 for 1 h. (**D**) RBL-2H3 cells were degranulated by compound 48/80 for 1 h. The amount of β-hexosaminidase secreted was evaluated using a β-hexosaminidase activity assay; (**E**) the GC-MS chromatogram of Gt-EE. A phytochemical fingerprinting profile of this extract was obtained by GC-MS analysis. (**A**–**D**) are presented as the mean ± standard deviation. ## *p* < 0.01, ### *p* < 0.001 compared with the normal group, and ** *p* < 0.01, *** *p* < 0.001 compared with the control group.

**Figure 2 plants-11-02540-f002:**
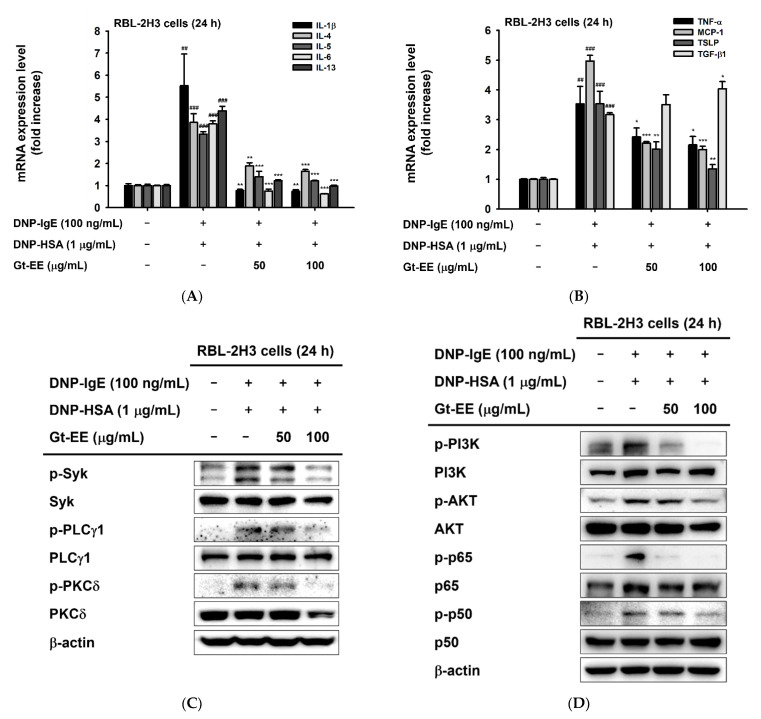
Gt-EE suppressed the mRNA expression of allergic cytokines and activation of the IgE–FcεRI signaling pathway. (**A**,**B**) The mRNA expression levels of IL-1β, IL-4, IL-5, IL-6, IL-13, TNF-α, MCP-1, TSLP, and TGF-β1 in IgE-stimulated RBL-2H3 cells were investigated using real-time PCR. IgE-sensitized RBL-2H3 cells were treated with Gt-EE for 30 min and then challenged with DNP-HAS for 24 h. (**C**–**E**) The total or phosphorylated forms of Syk, PLCγ1, PKCδ, PI3K, AKT, NF-κB p65, NF-κB p50, p38, JNK, and ERK1/2 in IgE-stimulated RBL-2H3 cells were detected using an immunoblotting analysis. IgE-sensitized RBL-2H3 cells were treated with Gt-EE for 30 min and then stimulated with DNP-HAS for 24 h. (**A**,**B**) The results are expressed as mean ± standard deviation. ## *p* < 0.01, ### *p* < 0.001 compared with the normal group, and * *p* < 0.05, ** *p* < 0.01, *** *p* < 0.001 compared with the control group.

**Figure 3 plants-11-02540-f003:**
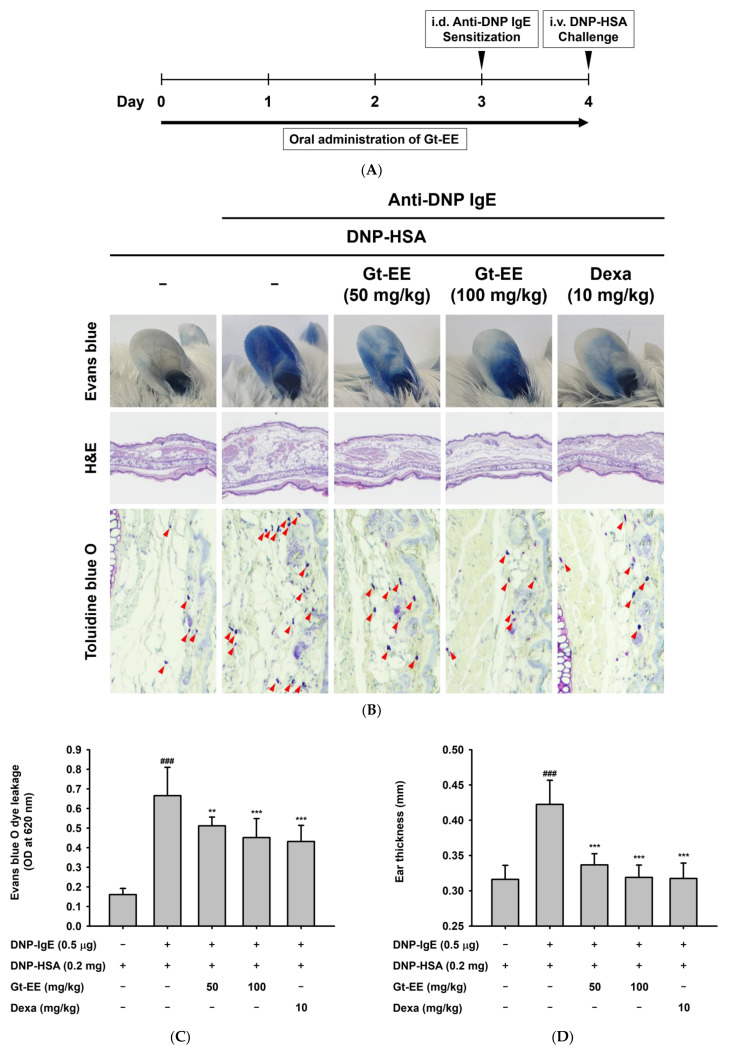
Effects of Gt-EE on PCA. (**A**) Experimental procedure for induction of PCA. Anti-DNP IgE and DNP-HSA were administered to the ears of BALB/c mice to induce PCA, and then the ears were stained with Evans blue dye. Gt-EE was orally administered every day for 5 days before PCA induction. Dexamethasone was used as the positive control. After euthanasia, the ears were dissected and soaked in formamide for extravasation of the Evans blue dye or fixed for histopathological analysis; (**B**) photographs of the Evans blue–stained ears and ear tissues stained with hematoxylin and eosin (H&E) or toluidine blue O. Histopathological variations and alterations in the mast cell counts due to IgE–antigen induction were assessed using H&E and toluidine blue O staining, respectively. The red arrows indicate toluidine blue O–stained mast cells; (**C**) absorbance of Evans blue extravasated from the mouse ears. Absorbance was measured at 620 nm. (**D**) The change in ear thickness was investigated using dial thickness gauges (PEACOCK, Japan). (**E**) The number of mast cells in the ear tissue was counted under a microscope at 400× magnification. (**F**) The mRNA expression of IL-4 in the mouse ears was measured using real-time PCR. After euthanasia, the ears were dissected and ground, and then total RNA was isolated for real-time PCR. (**C**–**F**) The results are presented as the mean ± standard deviation. ### *p* < 0.001 compared with the normal group, and * *p* < 0.05, ** *p* < 0.01, *** *p* < 0.001 compared with the control group.

**Figure 4 plants-11-02540-f004:**
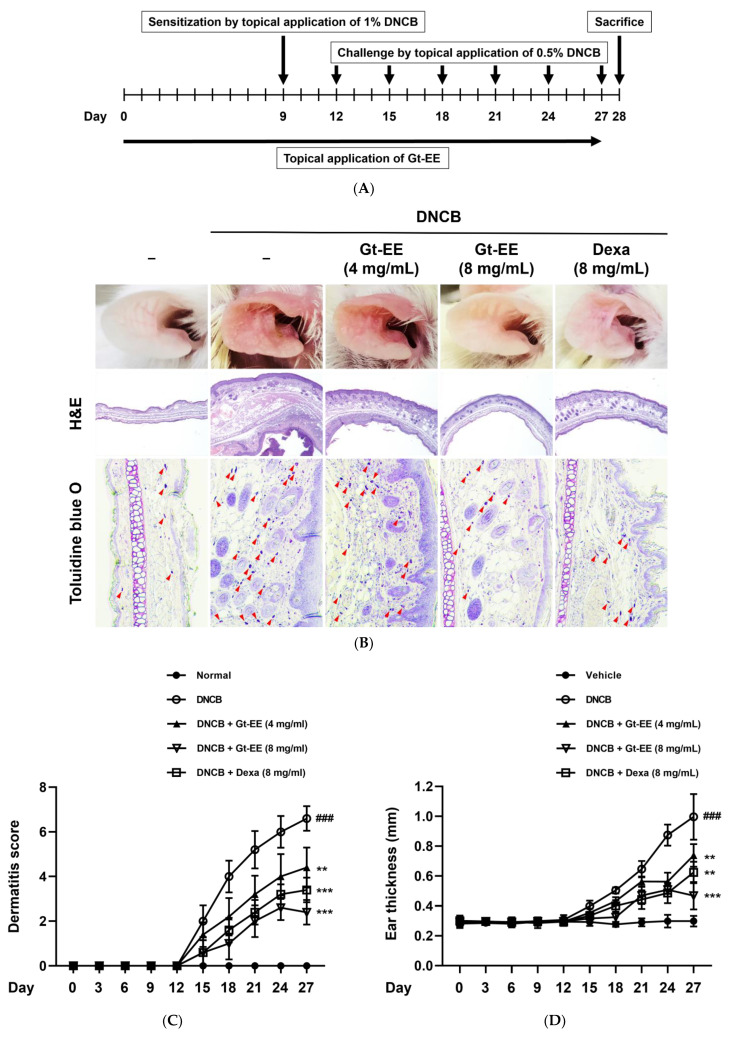
Effects of Gt-EE on DNCB-induced AD-like lesions. (**A**) scheme for induction of AD using DNCB in the ears of BALB/c mice. The ears were sensitized by 1% DNCB on day 9, and then 0.5% DNCB was applied to the ears every 3 days after sensitization. Gt-EE and dexamethasone were applied to the ears every day. Dexamethasone was used as the positive control; (**B**) photographs of BALB/c ears on which AD was induced and H&E- or toluidine blue O–stained ear tissues. Histopathological variations and alterations in mast cell counts due to DNCB treatment were investigated using H&E and toluidine blue O staining, respectively. The red arrows indicate toluidine blue O–stained mast cells. (**C**) The dermatitis score was measured every 3 days using itching, erythema, and dryness (No change—0, slight change—2, moderate change—4, severe change—8). (**D**) Alterations in ear thickness were measured using dial thickness gauges (PEACOCK, Japan). (**E**) Water content was evaluated using an SK-IV digital moisture monitor for skin (Pandawill, China) every 3 days. (**F**) The body weight of the mice was measured to check the toxicity of Gt-EE. (**G**) The number of mast cells in ear tissue was counted under a microscope at 400× magnification. (**H**) The serum IgE level was measured using ELISA. After euthanasia, blood was extracted directly from the heart, and the serum was separated by centrifugation. (**C**–**H**) The results are expressed as mean ± standard deviation. ### *p* < 0.001 compared with the normal group, and ** *p* < 0.01, *** *p* < 0.001 compared with the control group.

**Figure 5 plants-11-02540-f005:**
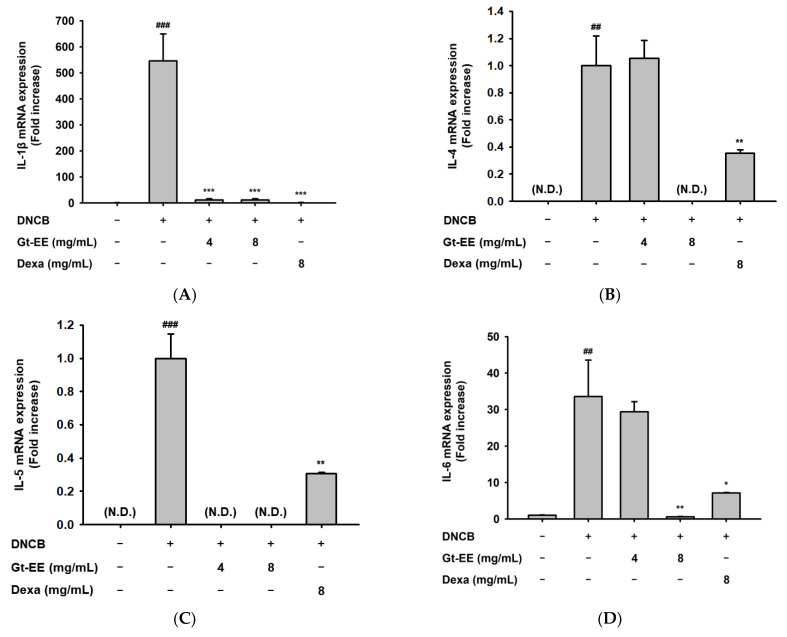
Gt-EE inhibits the mRNA expression of cytokines and activation of signaling pathways in DNCB-induced AD. (**A**–E) The mRNA expression levels of IL-1β, IL-4, IL-5, IL-6, and TNF-α were evaluated using real-time PCR. After euthanasia, the ears in which AD had been induced were dissected and ground, and total RNA was extracted for real-time PCR. (**F**,**G**) The total or phosphorylated forms of IκBα, NF-κB p65, p38, JNK, and ERK1/2 were measured using immunoblotting analysis. The total proteins were isolated from ears in which AD had been induced. (**A**–**E**) The results are expressed as mean ± standard deviation. Fold increase in 5A, 5D, and 5E was relative value compared with normal group, while fold increase in 5B and 5C was relative value compared with DNCB-treated group (Control group). N.D.: Not detected. ## *p* < 0.01, ### *p* < 0.001 compared with the normal group, and * *p* < 0.05, ** *p* < 0.01, *** *p* < 0.001 compared with the control group.

**Table 1 plants-11-02540-t001:** GC-MS phytochemical analysis of an ethanol extract of *Grewia tomentosa* Juss.

Peak No.	RT *	Name of the Compound	Corr. Area	Peak Area %
1	1.739	Acetic acid	24884923	4.330
2	5.208	dl-Threitol	18836124	3.278
3	8.351	Catechol	9546314	1.661
4	8.632	Benzofuran, 2,3-dihydro-	9566333	1.665
5	10.044	2-Methoxy-4-vinylphenol	8342175	1.452
6	10.530	Phenol, 2,6-dimethoxy-	11035072	1.920
7	13.191	Vanillic acid	7966671	1.386
8	14.262	3,4-Dimethylbenzyl isothiocyanate	21642735	3.766
9	15.222	4-((1E)-3-Hydroxy-1-propenyl)-2-methoxyphenol	21823662	3.798
10	15.618	6-Hydroxy-4,4,7a-trimethyl-5,6,7,7a-tetrahydrobenzofuran-2(4H)-one	13248125	2.305
11	17.430	n-Hexadecanoic acid	73947344	12.868
12	17.760	Hexadecanoic acid, ethyl ester	33280410	5.791
13	18.909	Phytol	15818890	2.753
14	19.120	9-Octadecenoic acid, (E)-	54693843	9.517
15	19.399	Ethyl Oleate	37006716	6.440
16	19.614	Octadecanoic acid, ethyl ester	5038345	0.877
17	20.149	Cyclopentanecarboxylic acid, 1-(2-butenyl)-2-oxo-, ethyl ester, (E)-	4901576	0.853
18	20.427	2-Amino-3-cyano-5,6-dimethoxy-1H-indenone	3073809	0.535
19	21.099	9-Octadecenamide, (Z)-	11696305	2.035
20	22.269	Hexadecanoic acid, 2-hydroxy-1-(hydroxymethyl)ethyl ester	3834019	0.667
21	23.493	(E)-3,3’-Dimethoxy-4,4’-dihydroxystilbene	3465908	0.603
22	23.666	1,3,12-Nonadecatriene	5898982	1.026
23	24.665	(1R,2R,4S)-2-(6-Chloropyridin-3-yl)-7-azabicyclo [2.2.1]heptane	1883262	0.328
24	28.303	Benzo[h]quinoline, 2,4-dimethyl-	5613204	0.977
25	28.669	N-Methyl-1-adamantaneacetamide	8297566	1.444
26	29.385	γ-Sitosterol	49971390	8.695
27	29.957	β-Amyrin	11442470	1.991
28	30.257	Arsenous acid, tris(trimethylsilyl) ester	4933525	0.858
29	30.586	Lupeol	35435894	6.166
30	31.338	1,2,5-Oxadiazol-3-amine, 4-(4-methoxyphenoxy)-	10571767	1.840
31	31.740	Tetrasiloxane, decamethyl-	5391505	0.938
32	32.347	1,2-Bis(trimethylsilyl)benzene	7023304	1.222
33	32.803	Friedelan-3-one	34569990	6.015

* Retention time (min).

**Table 2 plants-11-02540-t002:** Inhibition of 15-lipoxygenase activity by Gt-EE.

Sample (μg/mL)		% Inhibition
Gt-EE	50	34.6 ± 3.5
Gt-EE	100	43.6 ± 4.0
Quercetin	10	58.3 ± 6.6

**Table 3 plants-11-02540-t003:** Sequences of primers used for PCR.

Gene Name	Sequence (5′–3′)	
IL-1β	Forward	AGGCTGACAGACCCCAAAAG
(Rat)	Reverse	CTCCACGGGCAAGACATAGG
IL-4	Forward	TGTACCGGGAACGGTATCCA
(Rat)	Reverse	ACATCTCGGTGCATGGAGTC
IL-5	Forward	AGAATCAAACTGTCCGAGGGG
(Rat)	Reverse	ACTCATCACGCCAAGGAACTC
IL-6	Forward	ACAAGTCCGGAGAGGAGACT
(Rat)	Reverse	TTCTGACAGTGCATCATCGC
IL-13	Forward	GCTCTCGCTTGCCTTGGTGG
(Rat)	Reverse	CATCCGAGGCCTTTTGGTTA
TNF-α	Forward	AGATGTGGAACTGGCAGAGG
(Rat)	Reverse	CCCATTTGGGAACTTCTCCT
MCP-1	Forward	AGCCAACTCTCACTGAAGCC
(Rat)	Reverse	AACTGTGAACAACAGGCCCA
TSLP	Forward	TCAGGCAACAGCATGGTTCT
(Rat)	Reverse	AAGTTAGTGCCAGCCGTACC
TGF-β1	Forward	TGACGTCACTGGAGTTGTCC
(Rat)	Reverse	GTGAGCACTGAAGCGAAAGC
β-actin	Forward	TAACCAACTGGGACGATATG
(Rat)	Reverse	ATACAGGGACAGCACAGCCT
IL-1β	Forward	GCCCATCCTCTGTGACTCAT
(Mouse)	Reverse	AGGCCACAGGTATTTTGTCG
IL-4	Forward	ACAGGAGAAGGGACGCCAT
(Mouse)	Reverse	GAAGCCCTACAGACGAGCTCA
IL-5	Forward	CTCTGTTGACAAGCAATGAGACG
(Mouse)	Reverse	TCTTCAGTATGTCTAGCCCCTG
IL-6	Forward	AGCCAGAGTCCTTCAGAGAGAT
(Mouse)	Reverse	AGGAGAGCATTGGAAATTGGGG
TNF-α	Forward	TGCCTATGTCTCAGCCTCTT
(Mouse)	Reverse	GAGGCCATTTGGGAACTTCT
GAPDH	Forward	TGTGAACGGATTTGGCCGTA
(Mouse)	Reverse	ACTGTGCCGTTGAATTTGCC

## Data Availability

The data used to support the findings of this study are available from the corresponding author upon request.

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
