# Peer review of "Inhibitory Effects of Grewia tomentosa Juss. on IgE-Mediated Allergic Reaction and DNCB-Induced Atopic Dermatitis"

_plants, 2022, doi:10.3390/plants11192540_

Round 1
Reviewer 1 Report
In this manuscript entitled “Inhibitory Effects of Grewia tomentosa Juss. on IgE-mediated Allergic reaction and DNCB-induced atopic dermatitis” by Lee et al., the authors described the anti-inflammatory activity of G. tomentosa ethanol extract (Gt-EE). Gt-EE had anti-allergic effect in both in vitro and in vivo mouse model. Gt-EE also ameliorated symptoms of DNCB-induced atopic dermatitis.
In this manuscript, the authors drew appropriate conclusions through experiments utilizing in vitro and in vivo model systems suitable for allergy and atopic symptoms. It is recommended to publish this manuscript in this journal is as all strategies and experiments are reasonable and the results are clear with exception of minor points.
Minor points
1. Explain Fig. 1E on the legend of Figure 1. Otherwise, Fig.1E does not match the figure title, I recommend moving it to supplementary data.
2. Page 4, line 131, Figure 1F should be changed to Figure 1E.
3. Contents of Table 2 should be improved (correction of bold characters and position of line)
4. Page 6, lines 165-167, what is(are) the possible explanation(s) for the increased TGF-betta1 by Gt-EE treatment?
5. In Fig.5, the title ‘fold increase’ on the y-axis is unclear. ‘Fold increase’ compared what? Gapdh? or DNCB-treated group? Are statistical analyzes meaningful with ‘N.D’ values as control (Fig. 5B and C)?
6. In Fig. 5G, what is the possible reason(s) for phospho-p38 to increase after treatment of Gt-EE in the AD model, despite the same treatment decreased in vitro (Fig. 2E)?
Reviewer 2 Report
The manuscript focuses on a topic relevant to the scientific community. The experiments performed are well structured and the results obtained support the conclusions presented. However, since the plant species has been little studied and there are no other manuscripts in the literature, it is important to detail the methodology for obtaining the extract in the material and methods section. It will also be important to explain the choice of concentrations of the extract used in the various experiments.
